# Biophysical Evaluation of Water-Soluble Curcumin Encapsulated in β-Cyclodextrins on Colorectal Cancer Cells

**DOI:** 10.3390/ijms232112866

**Published:** 2022-10-25

**Authors:** Zhi Xuan Low, Michelle Yee Mun Teo, Fariza Juliana Nordin, Firli Rahmah Primula Dewi, Vijayaraj Kumar Palanirajan, Lionel Lian Aun In

**Affiliations:** 1Department of Biotechnology, Faculty of Applied Sciences, UCSI University, Kuala Lumpur 56000, Malaysia; 2Department of Biology, Faculty of Science and Technology, Universitas Airlangga, Surabaya 60115, Indonesia; 3Department of Pharmaceutical Technology, Faculty of Pharmaceutical Sciences, UCSI University, Kuala Lumpur 56000, Malaysia

**Keywords:** cyclodextrin, curcumin, inclusion complex, colorectal cancer, solubility, anticancer, encapsulation

## Abstract

Curcumin (CUR), a curcuminoid originating from turmeric root, possesses diverse pharmacological applications, including potent anticancer properties. However, the use of this efficacious agent in cancer therapy has been limited due to low water solubility and poor bioavailability. To overcome these problems, a drug delivery system was established as an excipient allowing improved dispersion in aqueous media coupled with enhanced in vitro anticancer effects. Different analyses such as UV–vis spectroscopy, differential scanning calorimetry (DSC), Fourier transform infrared spectroscopy (FTIR), scanning electron microscopy (SEM), solubility and dissolution assays were determined to monitor the successful encapsulation of CUR within the inner cavity of a β-cyclodextrin (β-CD) complex. The results indicated that water solubility was improved by 205.75-fold compared to pure CUR. Based on cytotoxicity data obtained from MTT assays, the inclusion complex exhibited a greater decrease in cancer cell viability compared to pure CUR. Moreover, cancer cell migration rates were decreased by 75.5% and 38.92%, invasion rates were decreased by 37.7% and 35.7%, while apoptosis rates were increased by 26.3% and 14.2%, and both caused caspase 3 activation toward colorectal cancer cells (SW480 and HCT116 cells). This efficacious formulation that enables improved aqueous dispersion is potentially useful and can be extended for various chemotherapeutic applications. Preliminary toxicity evaluation also indicated that its composition can be safely used in humans for cancer therapy.

## 1. Introduction

An increasing trend in the combination of conventional chemotherapeutics with native perennial herb medicine has been reported to show potent phytoconstituents with a wide range of biological activity [1,2,3,4,5]. Unfortunately, the development of many of these anticancer phytocompounds is often obstructed by efficacy issues owing to poor solubility and bioavailability [6], tumor heterogeneity, dose-limiting toxicity, low therapeutic indices, and drug resistance [7,8]. Therefore, improved formulation approaches in its drug delivery system are essential to bolster synergistic antitumor efficacies and clinical safety before incorporating them into chemotherapy regimens.

Turmeric (*Curcuma longa* Linn.) belongs to the ginger family, Zingiberaceae, and indicates that curcumin (CUR) is one such remedy indigenous to the Southeast Asian continent. The yellow phenolic agent is a principal curcuminoid compound containing secondary metabolites that exhibit powerful advantages over traditional chemotherapeutic multidrug, including broad antitumor properties and low toxicity outcomes [9,10,11,12,13]. Despite its extensive pharmacological activities against multiple diseases, the low bioavailability and poor solubility of natural CUR have limited its chemical application, which corresponds to the pharmacokinetic restriction of these molecules [14,15,16,17,18]. To further address the issues of rapid metabolism and limited absorption in the dissolution rate, encapsulation-based formulations have been generated.

Βeta cyclodextrin (βCD) is a delivery vector with hydrophilic hydroxyl groups on the outside and a hydrophobic cavity on the inside that can act as a nonpolar matrix. It is selected among a wide spectrum of carriers in materials science and pharmaceutics, particularly based on their capabilities to effectively entrap drug molecules and control drug release regarding their unique properties. Generally, complexation with βCD can lead to an alternation in the physicochemical properties and beneficial modifications of guest molecules by increasing their capacity to functionalize the products and enhance their affinity in water toward those temporarily locked within the host cavity due to its dimensional and geometrically limited fit characteristics [19,20,21,22,23].

This study was aimed at enhancing the solubility issue of CUR through interaction with βCD complexation and by investigating its ameliorated cytotoxicity and apoptotic effects against colorectal cancer (CRC) as a model. Nevertheless, the inclusion complex is a novel drug delivery system that can potentially improve the biological activities of CUR in different clinical and health applications.

## 2. Results

### 2.1. Physicochemical Characterization of the CUR Inclusion Complex with βCD

In this study, the supramolecular chemistry approach led to the generation of stable βCD-CUR inclusion complex (light yellow color fluffy powder) from the CUR (orange powder) and βCD (white powder) parent compounds. The absorbance spectra of CUR, βCD, and βCD-CUR inclusion complexes were recorded to evaluate the host-guest interaction between βCD and CUR (Figure 1a). The obtained result showed that βCD had no absorption peak in the range of 300–500 nm, while the typical maximum absorption peak (λ_max_) of CUR was observed at 430 nm. The βCD-CUR inclusion complex revealed a sharp peak at 350 nm and a weak peak at 442 nm. The peak belonging to CUR was not detected in the βCD-CUR inclusion complex, thus confirming that CUR was successfully loaded into the delivery vector βCD.

The chemical absorption of the βCD-CUR inclusion complex was characterized by a Fourier transform infrared spectroscopy (FTIR) spectrophotometer. The FTIR spectra of CUR (red line), βCD (light blue line), and βCD-CUR inclusion complexes (purple line) were demonstrated in Figure 1b. The absorption spectrum of CUR exhibited a sharp peak at 3507 cm^−^^1^, indicating the presence of the phenolic O-H stretching vibration, and a broad peak at 1626 cm^−^^1^ corresponding to C=O conjugation. Additionally, stretch vibrations of the benzene ring and C=C vibrations of CUR were shown at 1507 cm^−^^1^. In addition, the FTIR spectrum of βCD indicated absorption peaks at 3281 cm^−^^1^ and 2910 cm^−^^1^ due to the O-H group and C-H stretching, respectively. Another peak at 1644 cm^−^^1^, 1152 cm^−^^1^, 1020 cm^−^^1^, and 855 cm^−^^1^ representing C=O, C-O, and C-O-C stretching variations were also seen. As expected, all the absorption of the βCD-CUR inclusion complex was identical with the βCD spectrum as it shifted to higher or lower wavenumbers, such as 3277 to 3281 cm^−^^1^, 2918 to 2910 cm^−^^1^, 1626 to 1644 cm^−^^1^, 1427 to 1416 cm^−^^1^, 1150 to 1152 cm^−^^1^, and 1022 to 1020 cm^−^^1^. All these results provide substantial evidence of the formation of an inclusion complex.

A DSC thermogram of CUR, βCD, and βCD-CUR is shown in Figure 1c. CUR exhibited an individual endothermic peak at 169.5 °C, which corresponded to its melting point. In contrast, βCD-CUR displayed two endothermic peaks at 232.2 °C and 115.7 °C, respectively, which corresponded to the endothermic peaks of CUR and βCD. The melting peak was almost absent, which exhibited a shift from the peak that corresponded to βCD. The formation of degradation ions in the region after 200 °C was also observed, which promoted greater thermal protection of CUR. This can be attributed to the DSC curve of the βCD-CUR inclusion complex showing the features of βCD, thereby concluding the strong interaction of βCD and CUR molecules in mixed states and the formation of the inclusion complex as confirmed by the occurrence of broadening peaks.

An SEM study was performed to illustrate the microscopic morphological structures of CUR, βCD, and βCD-CUR inclusion complexes, as shown in Figure 2. CUR (Figure 2a) was observed to have an irregular spherical shape, while βCD (Figure 2b) was shown to have a crystalline flake-like structure. In the case of the βCD-CUR inclusion complex (Figure 2c), it was seen as a morphological combination of the parent compounds by exhibiting a combination of large and small irregular-shaped clumps. This morphological change suggested the successful formation of an inclusion complex.

### 2.2. Inclusion Complex Aqueous Solubility and CUR Release Profile

The encapsulation efficiency of the βCD-CUR inclusion complex was calculated spectroscopically from the absorbency of the supernatant at 430 nm, which was approximately 23.97%. By contrast, the βCD-CUR inclusion complex was completely dissolved to form a well-dispersed solution, while CUR was insoluble in water. The solubility of the βCD-CUR inclusion complex was dramatically increased compared to CUR, from 0.4% to 82.3%. This subsequently confirmed the complete formation of an inclusion complex in this study.

The drug release profiles were examined for the CUR and βCD-CUR inclusion complexes at pH 1.5 and pH 6.8 buffer (Figure 3). The results showed that the exhibited dissolution rate in the βCD-CUR inclusion complexes was more rapid, as 44.78% drug release was achieved within 120 min, whereas the percentage of drug release from CUR was significantly lower (*p* < 0.00003) and only reached 10.94%. At 360 min, 53.13% and 96.71% of the CUR and βCD-CUR inclusion complexes were released, respectively. The cumulative drug release of CUR from the inclusion complex reached nearly 100% (*p* < 0.000004). The increase in the dissolution rate can be attributed to the fact that CUR was converted into an amorphous form.

### 2.3. Evaluation of the Antiproliferation Effect of the CUR Inclusion Complex

The antitumoral effect of the various treatments underlying the proliferation index was examined using the MTT assay. In this study, normal and cancer human cell lines were treated with several concentrations of samples over 24 h of incubation time. As shown in Figure 4, CUR and βCD-CUR inclusion complexes inhibited the growth of CRC cells (SW480 and HCT116) in a dose-dependent manner. Cytotoxic improvements were obtained in βCD-CUR inclusion complex treated with SW480 and HCT116 cell lines at IC_50_ values of 17.2 μM and 29.8 μM, in comparison to CUR at 25.5 μM and 33.1 μM, respectively (Figure 4a,b). No cytotoxicity effects were observed for βCD when treated with more than 250 μM. When tested with NHDF cells (Figure 4c), the cell viability of the βCD-CUR inclusion complex was above 88.4% (SI value > 10 for both SW480 and HCT116 cells), compared with CUR of 50.2% (SI value > 10 for SW480 cells and SI value > 3 for HCT116 cells), suggesting that the inclusion complex did not exhibit cytotoxicity against normal cells and had concealed CUR cytotoxicity at higher concentrations.

### 2.4. Treatments of the CUR Inclusion Complex Augments Anti-Migration and Anti-Invasion Effects

The wound healing assay was used to investigate the effect of the CUR and βCD-CUR inclusion complex on cell migration. As shown in Figure 5a, the images were taken at 100× magnification by viewing under an inverted microscope, and there was a difference in the edge closure speed of the wound in the CUR and βCD-CUR inclusion complex. Cell migration of the βCD-CUR inclusion complex on SW480 and HCT116 cell lines was reduced, with 8.54% and 21.11% of the area migrated compared to control (Figure 5b). The migration ability of both cell lines treated with CUR was slightly weaker as compared to the βCD-CUR inclusion complex, at 18.89% and 31.31%, respectively (Figure 5b).

The invasion capabilities of the CUR and βCD-CUR inclusion complex on CRC cells were examined using a Matrigel-coated polycarbonate filter in a Boyden chamber. The crystal violet dye used to stain the transmitting cell through the Matrigel was observed, as seen in Figure 6a. At 24 h, the cell invasion of CUR was different when compared with the βCD-CUR inclusion complex for SW480 and HCT116 cell lines (Figure 6b). Similar effects of cell invasion were observed on SW480 and HCT116 cells for the treatment of CUR (76.7% and 81.9%), where a significant reduction (*p* < 0.05 and *p* < 0.01) of βCD-CUR inclusion complex (52.5% and 57.1%) were observed. The organic solvent (DMSO) used in dissolving CUR was examined together with βCD, and the obtained results showed that both had not affected cell-penetrating efficiency. These data demonstrate that CUR entrapped in the βCD cavity has a significant inhibitory effect on cancer cell invasion.

### 2.5. Cell Apoptosis Effects on Phase Identification and PARP Expression

To confirm apoptosis cell death induced by the treatment of CUR and βCD-CUR inclusion complex, Annexin V-FITC/PI analysis was carried out by flow cytometry. As shown in Figure 7a, it could be seen that after SW480 cells were treated with CUR and βCD-CUR inclusion complex, the proportion of early/late apoptosis cell population was increased from 61.23% to 79.77%, compared to the viable cells (0.55%). In comparison with the CUR treatment, the βCD-CUR inclusion complex significantly increased (*p* < 0.05) the early/late cell apoptosis from 80.20% to 92.50% on HCT116 cells (Figure 7b). Camptothecin was used as a positive control, which resulted in 24.52% and 55.62%, respectively, of early/late apoptosis on SW480 and HCT116 cells.

The induction of apoptosis-mediated cell death was confirmed using proteolytic PARP cleavage assays where full-length PARP (116 kDa) was cleaved into a large 89 kDa fragment through caspase-3 activity (Figure 8). As seen in Figure 8a, the Western blot image was represented for SW480 cells, whereas HCT116 cells were shown in Figure 8b. β-actin was used as a loading control.

## 3. Discussion

In this study, major obstacles to CUR have been addressed, which were poor aqueous solubility and low bioavailability, which severely limited its chemotherapeutic applications. To achieve this goal, CUR was complexed with βCD in a 2:1 molar ratio. Researchers have demonstrated that the aromatic ring of CUR was suitable for entrapment within the inner cavity of βCD in this molar ratio according to the steric factors of forming inclusion and exhibiting the highest binding constant during the complexation [24].

After the preparation, the inclusion complex formulation was characterized for its physicochemical properties. A UV–visible analysis was carried out to evaluate the host-guest interaction between βCD and CUR. The sharp peak at 430 nm was the characteristic of CUR extremities due to its aromatic rings with their hydroxyl and ether groups [25,26]. In particular, the spectrum of the βCD-CUR inclusion complex showed the shift of the peak center and revealed two distinguished peaks at 350 nm and 442 nm. This indicated the partial shielding of the excitable electrons and chromophores of functionalized βCD and the presence of CUR in the final product [27,28]. To further confirm its encapsulation properties, the FTIR technique was employed. The data revealed that all the peaks belonging to βCD appeared and shifted to higher or lower frequencies during complexation, with a few of the CUR peaks visible, which was therefore rationalized as being indicative of complex formation [29,30]. However, these observations were inadequate to confirm the occurrence of supramolecular complexation. The collected βCD-CUR inclusion complex was subjected to analysis with a DSC thermogram. The DSC curve ascertained that the disappearance of the endothermic peak was mainly due to the physical change of amorphization and melting point. Several studies found that the shifted peaks were consistent [31,32]. The SEM results obtained were in suitable agreement with a few studies [16,29], which attributed to the fact that the pure CUR in a flake-like structure to smaller aggregates during encapsulation in the inner cavity of βCD.

In this study, excellent water solubility for the inclusion complex was obtained without using ethanol or DMSO to dissolve it [33]. The result demonstrated that the well-dispersed solution of the βCD-CUR inclusion complex was generated by the outer surface of βCD, which was covered with hydrophilic moieties. This subsequently entrapped the hydrophobic CUR within the inner surface, which turned out to be highly amorphous in nature. Additionally, this approach caused a reduction in biotransformation and hydrolysis of CUR, thereby showing advanced physiochemical properties that may improve anticancer efficiency compared with pure CUR [23,34]. This phenomenon was further confirmed by a dissolution study. βCD-CUR showed a high amount of drug release rather than pure CUR. Therefore, this feasible method has proved to have high in vitro stability at physiological pH conditions [35].

To date, there were no reports on the anticancer properties of this complex formulation both in vivo and in vitro in CRC. Pure CUR has shown poor bioavailability and rapid degradation that limits its clinical application. Recent studies have shown that antiproliferation activity is attributed to its ability to induce cell death in human cancer cells [36]. In this study, it was found that the βCD-CUR inclusion complex exhibited better cytotoxicity effects compared to pure CUR on both CRC cell lines. The results obtained have been normalized by the control group, DMSO solvent. Although the low concentration of DMSO (<0.5%) is less toxic on both cell lines, it should be taken into consideration when used in humans [37].

This study also reported that the inclusion complex exhibited a greater inhibitory effect in both migration and invasion assays. In this case, apoptosis properties were investigated for their superior effects on cellular and molecular pathways. Numerous studies have demonstrated the potential of CUR in apoptosis induction in both the intrinsic and extrinsic apoptotic pathways, while the complexation requires a lower concentration in a shorter period to achieve a similar effect. These data also indicated the activation of caspase-3 in the direct role of proteolytic cleavage of cellular proteins throughout the expression of the PARP enzymes involved in several progressions of programmed cell death [38,39,40].

All the evaluation data in this study confirmed that high soluble formulations of CUR by βCD had a reasonable effect when it comes to enhancing CUR delivery in human absorption while improving CUR anticancer effects during complexation compared to pure CUR.

## 4. Materials and Methods

### 4.1. Formulation of an β-Cyclodextrin-Curcumin Complex (βCD-CUR)

CUR (cat. no. C7090, 95% purity, MW 368.38 g/mol) was obtained from Solarbio (Beijing, China), and βCD (cat. no. C8510; >97% purity, MW 1135.0 g/mol) was purchased from Sigma-Aldrich (St. Louis, MO, USA). The βCD-CUR inclusion complex was prepared in a 2:1 M ratio. βCD was completely dissolved in deionized water (dH_2_O) with gentle agitation and mixed with CUR in 99.7% (*v*/*v*) anhydrous ethanol in a closed dark glass container. The glass container was vigorously agitated for 4 h with the cap open to allow ethanol evaporation. Afterward, the solution was stirred for 5 h after cooling down to room temperature and stored overnight at 4 °C. The next day, the solution was centrifuged at 1000× *g* for 15 min and filtered through a 0.45 μm membrane filter. The filtrate was evaporated under reduced pressure, followed by drying in an oven (60 °C) to obtain the final inclusion complex, which was stored for subsequent measurements.

### 4.2. Absorption Spectra

A solution was prepared for βCD, pure CUR, and βCD: CUR inclusion complex, respectively, and scanned at a wavelength of 300 to 500 nm using FLUOstar Omega UV/Vis spectrometry (BMG Labtech, Ortenberg, Germany) to obtain the wavelength of maximum absorption (λ_max_).

### 4.3. Fourier Transform Infrared (FTIR) Spectrum

FTIR spectra were detected using Nicolet™ Summit Spectrometer (Thermo Fisher Scientific, Inc., Waltham, MA, USA) using OMNIC software. These samples were grounded in a spectroscopic grade potassium bromide (KBr) disc container, with a scanning range used from 4000 to 650 cm^−^^1^ to perform the measurements.

### 4.4. Differential Scanning Calorimetry (DSC)

The samples were sent to the Institute of Nanoscience and Nanotechnology, Universiti Putra Malaysia (UPM), for DSC analysis. Thermal analyses were measured by a DSC822e differential scanning calorimeter (Mettler Toledo, Columbus, OH, USA). This sample was heated from 35 to 300 °C at a heat-ramping rate of 10 °C/min under an ultrahigh nitrogen atmosphere with a flow rate of 50 mL/min.

### 4.5. Scanning Electron Microscopy (SEM)

The samples were sent to the Microscopy Unit, Institute of Bioscience, UPM, for imaging. SEM photographs were studied on a JSM-IT100 InTouchScope™ scanning electron microscope (JEOL SEM Technologies, Peabody, MA, USA). The dry samples were placed on carbon tape by coating them with a thin layer of gold at an excitation voltage of 20 kV.

### 4.6. CUR Content Analysis

Drug loading was carried out to determine the CUR content in the βCD-CUR inclusion complex. Briefly, 10 mg of βCD: CUR inclusion complex was dissolved in ethanol and gently shaken in the dark for 24 h at room temperature. To remove the clumps of βCD from the solution, the supernatant containing CUR was collected via centrifugation at 9300× *g* for 15 min and analyzed by FLUOstar Omega UV/Vis spectrometry (BMG Labtech, Germany) at 430 nm. Ultimately, a standard plot of pure CUR was prepared under the same conditions. The CUR quantification was calculated using the following equation: Inclusion complex formation efficiency (%) = mass of complexed CUR/total of CUR added theoretically × 100 [23].

### 4.7. Solubilization Test

The solubility of the βCD: CUR inclusion complex was studied in comparison with pure CUR by using FLUOstar Omega UV/Vis spectrometry (BMG Labtech, Germany). A total of 3.0 mg of βCD: CUR inclusion complex and 0.42 mg of pure CUR were dissolved in dH_2_O separately and centrifuged at 1600× *g* at 25 °C for 5 min. Then, the solution was filtered through a 0.45 μm filter membrane. The clear filtrate was collected, and each absorbance was recorded to determine the drug concentration (*n* = 6). All experiments were performed in three independent experiments [41].

### 4.8. Dissolution Study

A size-restricted dialysis tubing procedure was performed to measure the release of CUR from the complex by using a dialysis bag of molecular cut-off membrane tubes (MWCO 8–10 kDa, Sigma, Germany). The dialysis bag containing the drug solution was immersed in the beaker containing 50 mL dissolution medium with 0.5% sodium lauryl at pH 1.5 for 2 h. The temperature was controlled at 37 °C with the agitation set at 150 rpm under perfect sink conditions. The previous buffer was replaced by a pH 6.8 buffer, and the same procedure was repeated for 4 h. An aliquot was withdrawn from the external solution and replaced with the same volume of fresh dissolution medium after fixed time intervals. The sample drawn at each time point was filtered through a 0.45 µm pore size syringe filter and assayed for CUR concentration using UV/Vis spectrometry (BMG Labtech, Germany) at 430 nm. The measurements were repeated thrice [42].

### 4.9. Cell lines and Cell Culture

Human CRC cell lines HCT116 (ATCC^®®^ CCL-247™) and SW480 (ATCC^®®^ CCL-228™), as well as normal human cell control NHDF (ATCC^®®^ PCS-201-012™), were cultured in the media containing Dulbecco’s Modified Eagle’s Medium (DMEM; Gibco, Thermo Fisher Scientific, Inc., USA) supplemented with 100 U/mL penicillin-streptomycin (Gibco, Thermo Fisher Scientific, Inc., USA) and 10.0% (*v*/*v*) fetal bovine serum (FBS; Gibco, Thermo Fisher Scientific, Inc., USA). The cells were cultivated at 37 ^o^ C in humidified 95% air under a 5% CO_2_ atmosphere.

### 4.10. MTT Cell Viability Assay

Cytotoxic effects were determined by measuring MTT dye uptake and metabolism. An equal number of cells (1.0 × 10^4^ cells/well) were seeded in complete growth media. After overnight incubation at 37 °C, cells were treated with samples ranging from (2.7 to 268.5 µM). Positive control (1 µM camptothecin) and untreated control were included. Thereafter, an MTT dye reagent (0.5 mg/mL) was added to each well, and the plates were incubated in the dark at 37 °C for 2 h. The purple formazan crystals were dissolved in 100 μL of DMSO, and the absorbance was read at 570 nm using a FLUOstar Omega microplate reader (BMG Labtech, Germany). The values of IC_50_ were calculated using a plotted scattered graph, and the data were normalized to the cell number toward vehicle controls. The selectivity index (SI) value = (IC_50_ normal cell/IC_50_ cancer cell line) was calculated to evaluate the cytotoxicity of samples against non-malignant cell lines [43,44,45].

### 4.11. Cell Migration Analysis

Cells were seeded at a density of 1 × 10^5^ cells per well in complete growth media and incubated overnight at 37 °C with 5% CO_2_. Media containing 5 µg/mL of mitomycin-C (Sigma-Aldrich, St. Louis, MO, USA) was changed and incubated for an additional 2 h, then treated with IC_20_ of samples on cells. Uniform scratches were implemented on the cell-monolayer surface using a sterile pipette tip, while non-adherent cells were removed with 1x phosphate buffered saline (PBS). Then, cells were monitored at 0 h and 24 h, and the images of the scratched area were captured using a Nikon Eclipse TS100 inverted fluorescence microscope (Nikon Instruments, Fujisawa, Japan). Cell motility was measured by determining the distance between the wound boundaries using ImageJ v1.43 analysis software (NIH, Bethesda, MD, USA) [46,47].

### 4.12. Cellular Invasion Assay

This experiment was carried out according to the manufacturer’s protocol of BD BioCoat™ Matrigel TM Invasion Chamber. The 24-well transparent PET membrane 8.0 μM pore size inserts were coated with 40 μL of 1 mg/mL Matrigel (BD Biosciences, San Jose, CA, USA) and incubated at 37  °C. After 2 h, cells (1.0 × 10^5^ cells/well) were grown as monolayers by being loaded into the upper insert of the invasion chamber for 24 h. The cells were starved in serum-free media with samples, and the lower insert was loaded with complete growth medium as a chemo-attractant. The chamber was incubated overnight at 37 °C in 5% CO_2,_ and the invaded cells were fixed, stained, and eluted with acetic acid. Each transwell invasion membrane insert area was recorded by the absorbance intensity, and each experiment was performed three times [46].

### 4.13. Annexin V-FITC/PI Apoptosis Detection Assay

Cells (1 × 10^6^ cells/well) were harvested after treatment with samples using IC_50_. After 24 h, cells were collected and washed twice with cold 1x PBS and then resuspended in 100 μL of 1X binding buffer. Double staining with 5 µL Annexin V-FITC and 5 µL PI was added to the cell suspension and incubated in the dark at room temperature. The cell suspension was added with binding buffer until it reached 500 μL and analyzed using flow cytometry (BD FACS Caliber instrument, BD Biosciences, San Jose, CA, USA) [48].

### 4.14. Western Blot Analysis

Protein samples were extracted from cells (1.0 × 10^7^/mL) after treatment using the ProteinExt^®®^ mammalian nuclear and cytoplasmic protein extraction kit (TransGen Biotech Co., Ltd., Beijing, China). Total protein concentration was quantified and normalized using the Quick Start Bradford Protein Assay Kit 2 (Bio-Rad, Irvine, CA, USA) according to the manufacturer’s protocol. Protein samples were loaded into a sodium dodecyl sulfate (SDS) polyacrylamide gel and transferred to a polyvinylidene fluoride (PVDF) membrane (Thermo Fisher, Waltham, MA, USA) for Western blot analysis using a wet transfer system (Bio-Rad, Irvine, CA, USA). Then, incubation was performed with PARP 1 polyclonal antibody (1/1000) (Elabscience Biotechnology Inc., Houston, TX, USA) and beta-actin (1/2000) (Elabscience Biotechnology Inc., USA). Detection of bound antibodies was carried out using secondary HRP-conjugated goat anti-rabbit IgG (1/10,000) (Elabscience Biotechnology Inc., USA), and the bands were developed with TMB substrate (KPL, Gaithersburg, MD, USA) by viewing with a GS-800 Calibrated Imaging Densitometer (Bio-Rad, Irvine, CA, USA). The proteolytic cleavage of 116 kDa PARP into an 89 kDa fragment was assessed as the occurrence of apoptosis [47].

### 4.15. Statistical Analysis

Statistical analysis was determined using Microsoft Excel, and all data were expressed as mean  ±  S.D for at least three replicates. *p*-values were calculated using Student’s two-tailed *t*-test. *p* < 0.05 (*), *p* < 0.01 (**), and *p* < 0.001 (***) were considered as significant difference.

## 5. Conclusions

All evaluation data in this study have confirmed the successful formation of a high soluble βCD-CUR inclusion complex that enhanced the CUR delivery in human absorption while improving the anticancer effect of CUR during the complexation rather than pure CUR. Further in vivo studies and clinical trials for inclusion complex formulation targeting CRC cells are needed to prove the overall improved efficiency for colorectal malignancies and perhaps other malignancies.

## Figures and Tables

**Figure 1 ijms-23-12866-f001:**
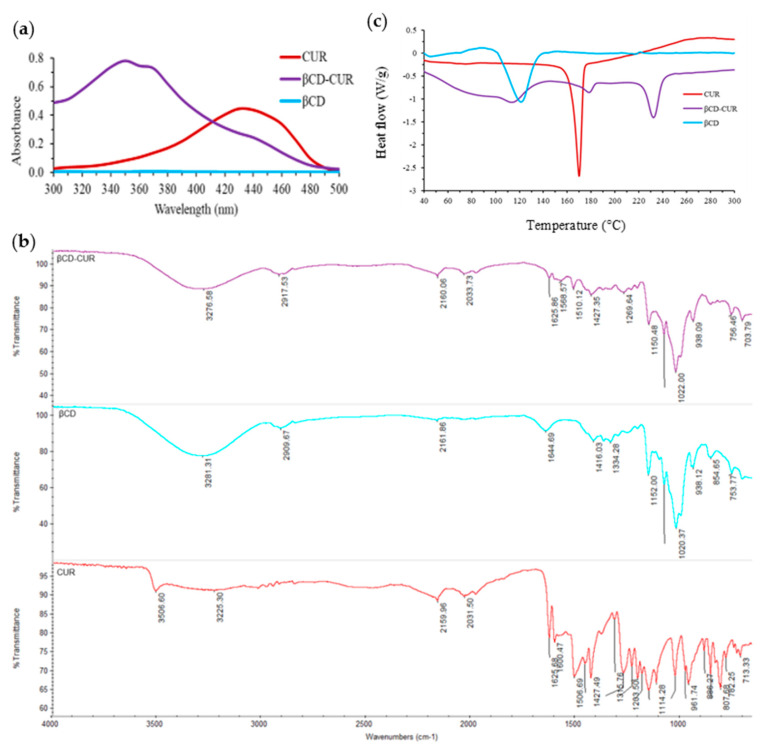
(**a**) Absorption spectra of CUR, βCD, and βCD-CUR inclusion complexes under a wavelength range of 300–500 nm. (**b**) Fourier transform infrared spectra (4000–1000 cm^−1^). (**c**) Differential thermal analysis exothermic curves.

**Figure 2 ijms-23-12866-f002:**
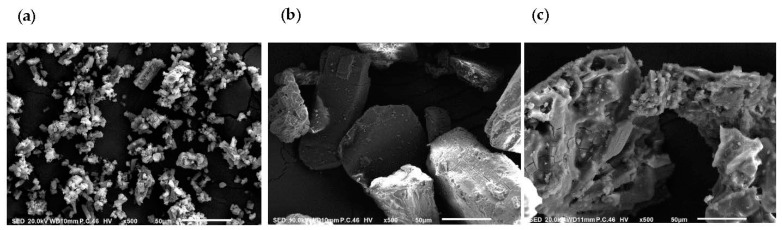
Scanning electron microscopy photomicrographs of (**a**) CUR, (**b**) βCD, and (**c**) βCD-CUR. The scale bars at 500× magnification were shown.

**Figure 3 ijms-23-12866-f003:**
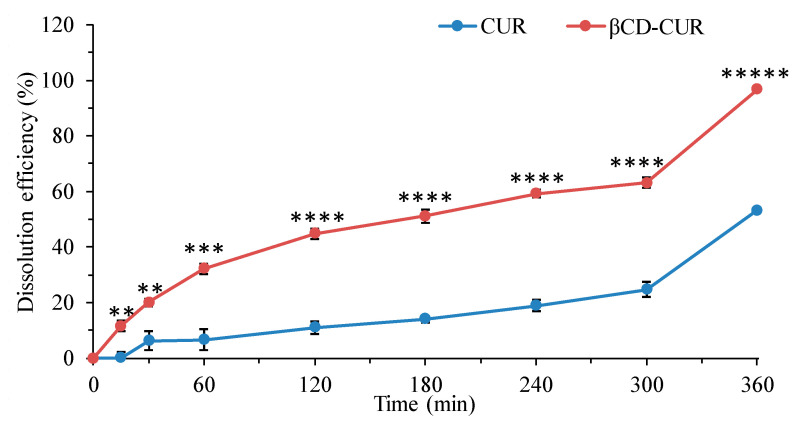
Time-dependent versus dissolution efficiency in two buffer media of pH 1.5 and pH 6.8 with 0.5% SLS. The release profile of CUR from the inclusion complex and pure CUR, ** *p* < 0.01, *** *p* < 0.001, **** *p* < 0.0001, and ***** *p* < 0.00001.

**Figure 4 ijms-23-12866-f004:**
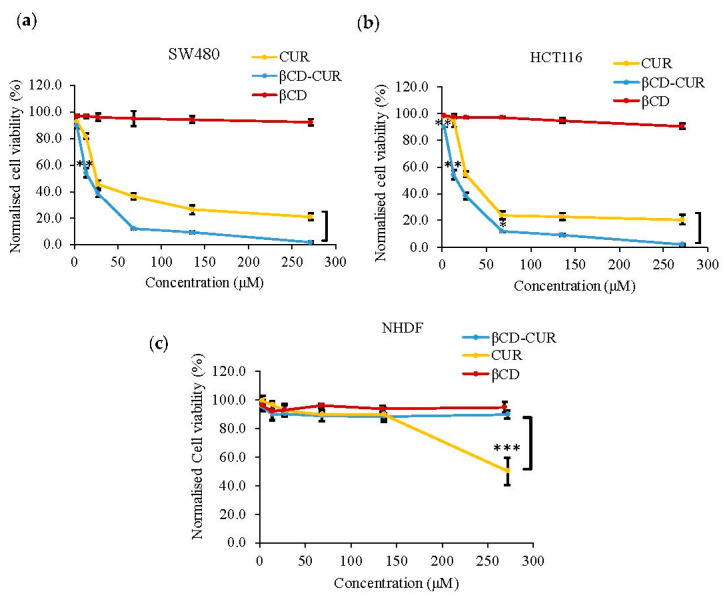
In vitro combined cytotoxic effects of βCD, CUR, and βCD-CUR. Normalized cell viability (%) after 24 h treatment against (**a**) SW480, (**b**) HCT116, and (**c**) NHDF cells. Data are shown as mean ± S.D. of three independent replicates, where statistically significant differences between CUR values versus βCD-CUR values are denoted as *p* < 0.05 (*), *p* < 0.01 (**), and *p* < 0.001 (***).

**Figure 5 ijms-23-12866-f005:**
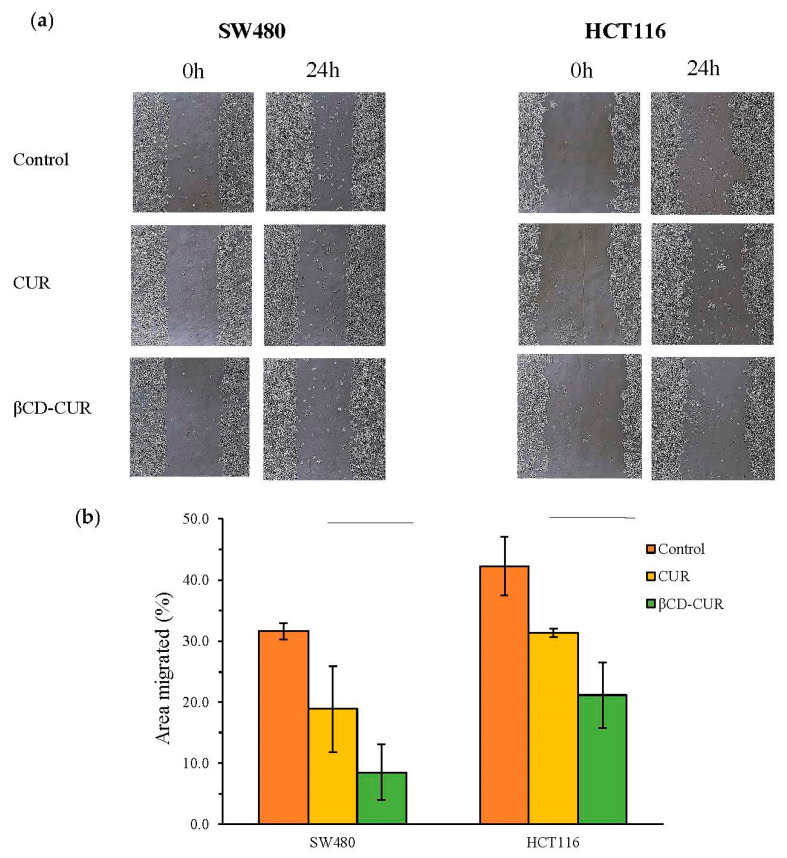
The effect of CUR and βCD-CUR inclusion complex on the migration properties of CRC cell lines. (**a**) The wound healing assays for SW480 and HCT116 cells. (**b**) The bar chart represents the measurement of the distance between the boundaries of the migrating cells 24 h after the treatment. The area migrated between CUR and βCD-CUR groups was reported as mean ± S.D. from three replicates.

**Figure 6 ijms-23-12866-f006:**
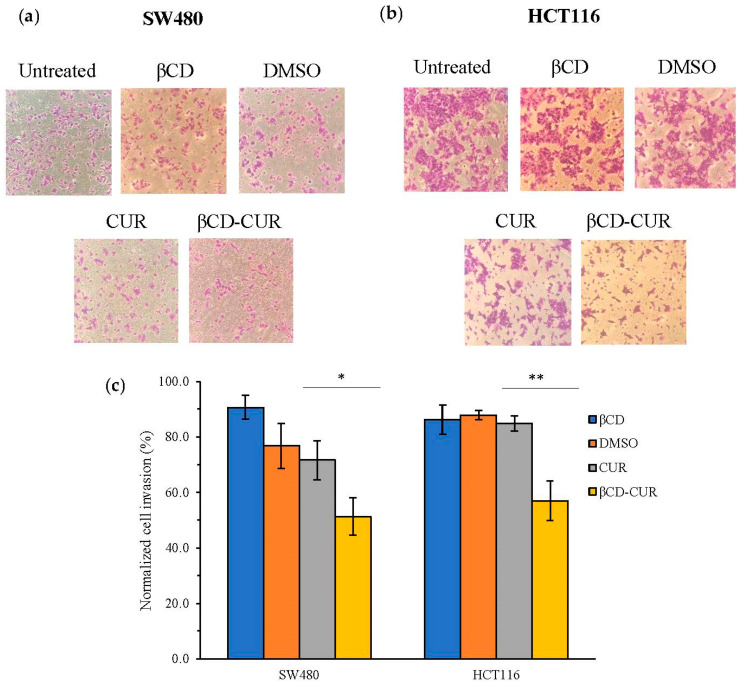
The inhibition effects on cell invasion as assessed by transwell assay. Representative images of invaded cells stained with crystal violet on the bottom membrane of Matrigel transwell invasion insert for (**a**) SW480 cells and (**b**) HCT116 cells at 200× magnification. (**c**) Bar graphs represent the normalized average number of invaded cells per field of SW480 and HCT116 cells. Quantified data were expressed as the mean ± S.D. from three independent experiments. Statistically significant differences in comparison between CUR and βCD-CUR are denoted with a *p* < 0.05 (*) and *p* < 0.01 (**).

**Figure 7 ijms-23-12866-f007:**
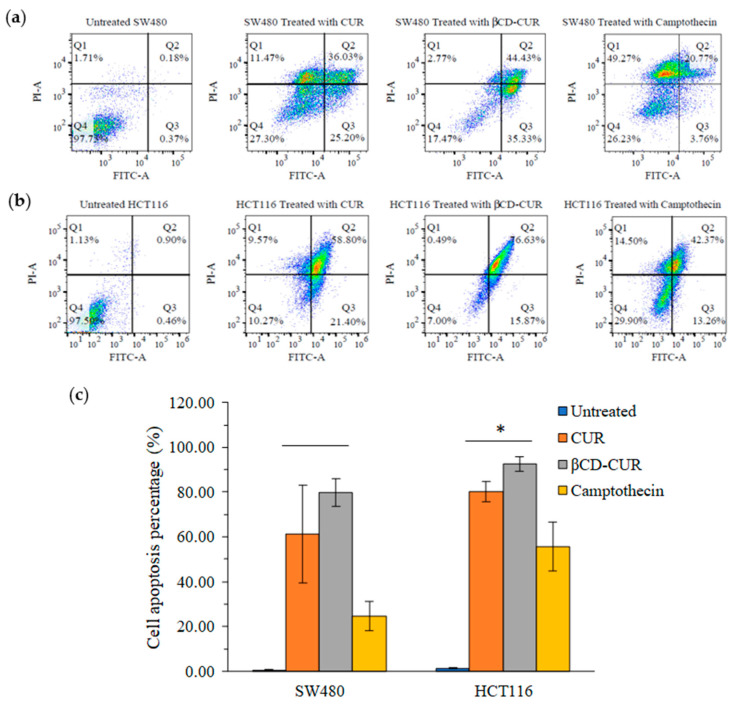
Annexin V/FITC-PI flow cytometry analysis of SW480 and HCT116 cells treated with negative control, CUR, βCD-CUR, and camptothecin for 24 h. Cells were dual stained with Annexin V-FITC (FITC) and propidium iodide (PI). (**a**) Dot plot of SW480 with different treatments. (**b**) Dot plot of HCT116 cells with different treatments. Each set of data shown was a representative plot of three independent experiments, while percentages were the mean value of three independent experiments. Q1, necrosis; Q2, late apoptosis; Q3, early apoptosis; Q4, viable cell. (**c**) The quantification of apoptotic rate on both cell lines. Data are presented as the mean ± S.D. (*n* = 3). * Significantly different, *p* < 0.05.

**Figure 8 ijms-23-12866-f008:**
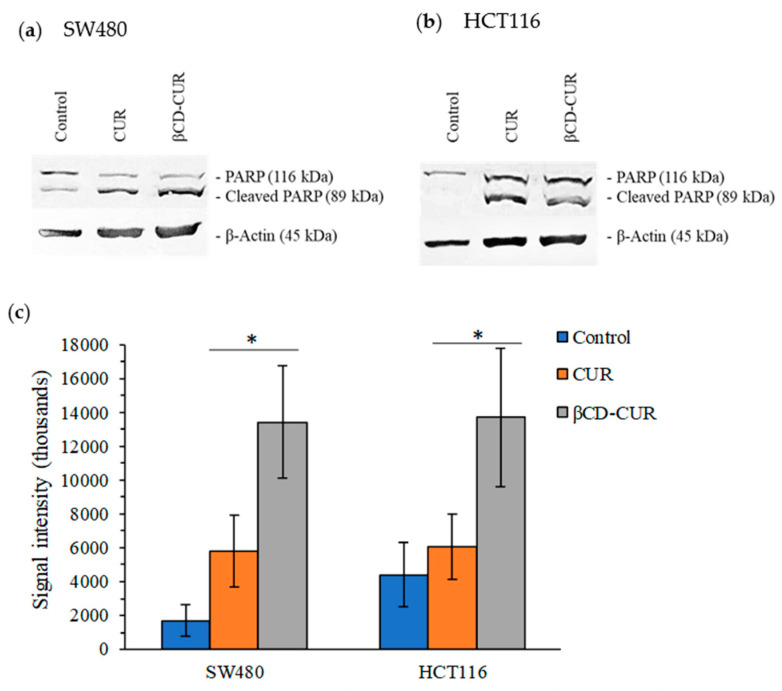
Induction of apoptosis through the activation of caspase-3 and subsequent cleavage of full-length PARP enzymes (116 kDa) into a large (89 kDa) subunit protein, with β-actin as a loading control for (**a**) SW480 cells and (**b**) HCT116 cells. (**c**) Cleaved PARP normalized to β-actin. Data for cleaved PARP were presented as mean ± S.D. of three independent replicates. Statistically significant changes between CUR and βCD-CUR groups are denoted as (*) with a *p* < 0.05.

## Data Availability

Not applicable.

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
