# Peer review of "Biophysical Evaluation of Water-Soluble Curcumin Encapsulated in β-Cyclodextrins on Colorectal Cancer Cells"

_ijms, 2022, doi:10.3390/ijms232112866_

Round 1

Reviewer 1 Report

In this study, the authors have constructed a new curcumin delivery formulation that greatly improves the water solubility of curcumin, which is an interesting study, and the results are promising. However, the experiments in terms of biological activity have many flaws that need to be refined. The text of the manuscript also needs to be re-edited.

1. Figure 2. Scanning electron microscopy photomicrographs were missing.

2. The results for cell migration in Figure 5 are less convincing; please confirm that the images in a and the data in b correspond.

3. Figure 6. "The inhibition effects on cell invasion as assessed by Transwell assay" was missing.

4. The flow cytometry results in Figure 7 are not satisfactory and there may be incorrect parameter settings, resulting in cell populations that are not clearly distinguishable. The cross gates were not positioned accurately.

5. In Figure 8, the label for b is missing. The protein immunoblotting experiment required three experiments to be repeated, and the authors only provided raw data for one experiment, requiring two additional experimental data.

Reviewer 2 Report

The authors did a great job showing that the βCD-CUR inclusion complex enables improved aqueous dispersion and is potentially helpful and can be extended for various chemotherapeutic applications. The introduction and material and methods sections are well written. However, I have a few comments that might be useful in improving the manuscript

1) Figure 5 has the name figure 6 and there is no figure 6 in the manuscript (the one for invasion assay)

2) In the migration assay, are you sure there is no significant difference between βCD-CUR and CUR with HCT116 cell line? the chart suggests there is a significant difference.

3) You must unify the sequence of the data presented in writing and in the figures. Sometimes you start with HCT116 cell line and at other times with SW480. This is so confusing.

4) In figure 7, the chart is not showing the same results as the dot plot with HCT116

5) I would suggest doing image analysis for the western plot bands since I do not see any difference by the eye and the analysis will show if there is any

6) In the MTT assay, How did you select the concentration? was it based on a pilot study or previous publication?

7) In the cellular invasion assay in the material and methods section, you should mention that the cells are added after the treatment

8) Did you use excel to do the statistical analysis (Student t-test)? or another statistical program?

Round 2

Reviewer 1 Report

The revision of the manuscript is satisfactory.